

**Biogeographical distribution of Microbial Communities along the Rajang River-South China**
**Sea Continuum**
Edwin Sien Aun **Sia**[1], Zhuoyi Zhu[2], Jing Zhang[2], Wee Cheah[3], Jiang Shan[2], Faddrine Holt Jang[1],
Aazani Mujahid[4], Fun-Kwo Shiah[5], Moritz Müller[1]
[1]Faculty of Computing, Engineering and Science, Swinburne University of Technology, Sarawak
Campus, Jalan Simpang Tiga, 93350, Kuching, Sarawak, Malaysia.
[2]State Key Laboratory of Estuarine and Coastal Research, East China Normal University, Zhongshan
N. Road 3663, Shanghai, 200062, China.
[3]Institute of Ocean and Earth Sciences, University of Malaya, 50603 Kuala Lumpur, Malaysia.
[4]Department of Aquatic Science, Faculty of Resource, Science and Technology, University Malaysia
Sarawak, 93400 Kota Samarahan, Sarawak, Malaysia.
[5]Research Center for Environmental Changes, Academia Sinica, Taipei 11529, Taiwan
Corresponding Author*: Moritz Müller, mmueller@swinburne.edu.my
**Abstract**
Microbial community composition and diversity in freshwater habitats, especially in lotic
environments, are much less studied compared to marine and soil communities. The Rajang River is
the main drainage system for central Sarawak in Malaysian Borneo and passes through peat domes
whereby peat-rich material is being fed into the system and eventually into the southern South China
Sea. Microbial communities found within peat-rich systems are important biogeochemical cyclers in
terms of methane and carbon dioxide sequestration. To address the critical lack of knowledge about
microbial communities in tropical (peat-draining) rivers, this study represents the first seasonal
assessment targeted at establishing a foundational understanding of the microbial communities of the
Rajang River-South China Sea continuum. This was carried out utilizing 16S rRNA gene amplicon
sequencing via Illumina MiSeq in size-fractionated samples (0.2 and 3.0 µm GF/C filter membranes)
covering different biogeographical features/sources from headwaters to coastal waters. The microbial
communities found along the Rajang river exhibited taxa common to rivers (i.e. the predominance of
*β-Proteobacteria*) while estuarine and marine regions exhibited taxa that were common to the
aforementioned regions as well (i.e. predominance of α- and γ-*Proteobacteria*). This is in agreement
with studies from other rivers which observed similar changes along the salinity gradients. In terms of
particulate versus free-living bacteria, nonmetric multi-dimensional scaling (NMDS) results showed
similarly distributed microbial communities with varying separation between seasons. Distinct
patterns were observed based on linear models as a result of the changes in salinity along with
variation of other biogeochemical parameters. Alpha diversity indices indicated that microbial
communities were higher in diversity upstream compared to the marine and estuarine regions whereas



anthropogenic perturbations led to increased richness but less diversity. Despite the observed changes
in bacterial community composition and diversity that occur along the Rajang River to sea continuum,
the PICRUST predictions showed minor variations. The results provide essential context for future
studies such as further analyses on the ecosystem health in response to anthropogenic land-use
practices and probable development of biomarkers to improve the monitoring of water quality in this
region.

Keywords: particle-associated microbes, free-living microbes, 16S rRNA, River-sea continuum









## 1.0 Introduction

Biogeochemical transformations are primarily governed by microbial communities (Konopka, 2009), and it is crucial to understand their dynamics in order to predict biosphere modulations in response to a changing climate. Despite the importance of freshwater to society and despite hosting the highest microbial diversity (Besemer et al., 2013), microbial community composition and diversity in freshwater habitats, especially in lotic environments, are much less studied compared to marine and soil communities (Kan, 2018).

Lotic environments are the interface between soil and aquatic environments and until not long ago, rivers were thought to be passive channels in the global and regional determination of carbon (C) and weathering products until it became clear that rivers regulate for example the transfer of nutrients from land to coastal areas (Smith and Hollibaugh, 1993). Several studies have shown that bacteria are key players in nutrient processing in freshwater systems (Cotner and Biddanda, 2002; Findlay, 2010; Madsen, 2011). Zhang et al. (2018a) stated that the organic matter composition is strongly modified by bacteria as well as its resistance to degradation. Bacteria strongly influence the fluvial organic matter, hence playing a role in carbon cycle (Dittmar et al., 2001) and recent studies in the Rajang river have demonstrated that as indicated by high concentrations of D-form amino acids (Zhu et al., 2019). Moreover, it was demonstrated by Jiang et al. (2019) that Dissolved Organic Nitrogen was reduced to $NH_4^+$ via mineralization and ammonification, again highlighting the biogeochemical activity and the importance of microbes in the Rajang River. Until now, there has, however, been no study on their diversity yet; a gap that this study aims to fill.

Next-generation sequencing technologies have enabled a better understanding of the rare or unculturable biosphere which traditional culture methods would not have been able to elucidate (Boughner and Singh, 2016; Cao et al., 2017). Only few studies assessing bacterial community composition have been undertaken in lotic/riverine environments (Fortunato et al., 2012; Ladau et al., 2013; Zwart et al., 2002), with even less focusing on the diversity of surface-attached biofilms in lotic environments, particularly in comparison to biofilm studies in benthic habitats (Zeglin, 2015). Furthermore, bacterial assemblages on suspended particles were shown to differ from free-living bacterioplankton in a number of studies (Bidle and Fletcher, 1995; Crump et al., 1999) in which the ratios between both fractions are often influenced by the quality of suspended particulate matter (Doxaran et al. 2012). Even less studies attempt to map bacterial community composition in a river-to-sea continuum across multiple seasons and habitats (Fortunato et al., 2012) and it was only recently reported that the most abundant riverine bacterioplankton resemble lake bacteria and can be regarded as 'typical' freshwater bacteria (Lozupone and Knight, 2007; Zwart et al., 2002). Metagenomics studies substantiated the dominance of *Proteobacteria* and *Actinobacteria* whereby *Bacteroidetes*, *Cyanobacteria*, and *Verrucomicrobia* were found also found to be abundant in rivers ((Cottrell et al.,





2005; Kolmakova et al., 2014; Lemke et al., 2009; Newton et al., 2011; Read et al., 2015; Staley et al.,
2013). While there are studies related to the freshwater-marine gradients of rivers such as studies by
Crump and Hobbie (2005) and Fortunato et al. (2013)and tropical peatlands (Kanokratana et al., 2011;
Mishra et al., 2014; Yule et al., 2016; Too et al., 2018), to the author's knowledge, this is the first
study which links both freshwater-marine gradients as well as tropical peatlands as a cohesive
component (i.e tropical peat-draining river to coastal ecosystem).  Due to the high diversity and fast
generation time, the first responders to environmental changes (both natural and anthropogenic events
such as storms, upwelling and pollutants) are microbial communities (Hunt and Ward, 2015). Liao et
al. (2019) show that extensive agricultural land-use in the inter-tidal region of a watershed resulted in
the prevalence of bacteria pathogen-like sequences whereas further Bruland et al. (2008)stated that the
assemblages of microbes also vary temporally as a function of oceanographic conditions, river
discharge, tidal phase and season. Thus, as the Rajang River experiences two monsoonal seasons
(Sa'adi et al., 2017) and is subjected to anthropogenic disturbances (Gaveau et al., 2016; Miettinen et
al., 2016), it is thus fundamental to take into consideration both seasonal and anthropogenic influences
on the microbial communities of the Rajang River.

Lotic environments are the interface between soil and aquatic environments as terrestrial
environments seed microbes into the adjacent water column due to flowing waters (Crump et al.,
2012).  Thus, it is essential to understand the dynamics and structure of microbial communities in
them to assess their contribution towards biogeochemical fluxes such as carbon and nitrogen (Battin
et al., 2008; Raymond et al., 2013), as well as phosphate cycling (Hall et al., 2013). In addition, the
fluxes as well as transformations of organic matter as well as nutrients in aquatic systems are
environmentally driven by parameters such as temperature or the availability of nutrients in these
ecosystems (Welti et al., 2017). In turn, various gradients (i.e physical, chemical, hydrological or even
biological) contribute to the changes in the microbial diversity and distribution living within the lotic
environments (Zeglin, 2015).

Given the rapid development in Sarawak and the hypothesized importance of microbes in several
biogeochemical processes in the Rajang river (Jiang et al., 2019; Martin et al., 2018; Müller-Dum et
al., 2019; Zhu et al. 2019), it is imperative to study the microbial communities to enable future
predictions and management responses. The Rajang river offers the opportunity to study the microbial
diversity along a river to sea continuum and at the same time assess influence of natural conditions
such as seasons (dry *vs*. wet), different soil types (peat *vs*. mineral soil), as well as anthropogenic
disturbances such as plantations. Linear models are used to examine the relationship between the
microbial community structure and their environment. This study aims to investigate (1) the microbial
community structure, diversity and probable function across wet and dry seasons in order to (2)





understand the underlying factors that may influence the spatial and seasonal distribution of the
prokaryotic communities and the nutrient dynamics involved in the Rajang River.


**2.0 Methodology**
**2.1 Study area and sampling strategy**
This study was conducted along ~300km of the Rajang river in Sarawak, Malaysia (Figure 1A). This
region has an equatorial climate characterized by constant temperatures, high extensive rainfall and
high humidity (Wang et al., 2009, 2005; see also **Supp. Fig. 1**). The Rajang delta system consists of
an alluvial valley, an associated coastal plain and a delta plain (Staub and Esterle 1993, pdf). The
coastal plain is dissected into several small distributaries, namely the Igan, Lassa, Paloh and Rajang
tributary (Fig. 1(A)). The shoreline experiences tides and seasonally strong waves ranging from 3 – 6
m with intensity increasing from the east to the west. According to Wetlands International (2015), the
land surrounding the study sites is characterised by a range of anthropogenic activities, ranging from
oil palm and sago plantations to human settlements as well as transportation and sand dredging
activities (Fig. 1(B)).

A total of 59 water samples were collected along salinity-gradients during three (3) cruises (**Fig. 1(A)**),
covering both wet and dry seasons as well as different source types (i.e. mineral or peat soils). Source
types sampled were grouped as follows: 1) marine 2) brackish peat 3) freshwater peat and 4) mineral
soils. From Sibu towards Kapit (upriver), the riparian zone is mineral soil whereas from Sibu
downwards to the coast it consists of peat which was then further divided into freshwater (salinity 0 to
~ 1 PSU) and brackish (salinity 2- 28 PSU)(as described in **Fig. 1(B)**).The cruise in August 2016
represented the highest sampling frequency in order to obtain complete coverage of representative
regions, while the cruises in March and September 2017 were aimed to obtain seasonal
representatives for each region. About 250 – 500 mL of water were filtered through 3.0 µm pore size
polycarbonate filters GF/C (Cyclopore, Whatman, Germany) via vacuum filtration. This was referred
to as the 'Particulate-attached' fraction. The filtrate from the 3.0 µm portion was collected in a sterile
glass bottle and subsequently filtered through 0.2 µm pore size polycarbonate (GF/C) filters
(Cyclopore, Whatman, Germany). The smaller fraction was referred to as 'free-living' fraction. All
filters (117 in total as 1  3.0 µm filter was contaminated and discarded during the filtration process)
were immediately stored at -20 °C and sent to the Australian Centre for Ecogenomics (ACE),
Brisbane for processing utilizing Illumina (Caporaso et al., 2012) platform.

**2.2 Pyrosequencing and Bioinformatics Analyses**
Initial upstream processes were carried out by the Australian Centre for Ecogenomics utilizing the
ACE mitag pipeline (ACE, 2016). In short, fastq files generated from the Illumina platform were





processed with fastqc, primer sequences trimmed with Trimmomatic, and poor quality sequences
removed using a sliding window of 4 bases with an average base quality of more than 15. Subsequent
processing steps were then performed utilizing the mothur pipeline. Sequences were aligned against
the SILVA alignment (Quast et al., 2013; Yilmaz et al., 2014), 'pre.cluster' command executed for
denoising, and chimeric sequences removed using the 'chimera.vsearch' function. Chimera-free 16s
rRNA bacterial gene sequences were taxonomically assigned against the EzTaxon database (Kim et
al., 2012) using the Naïve Bayesian classifier with a threshold of 80%. The quality-filtered sequences
were then clustered into operational taxonomic units (OTUs) at 97% similarity cutoff with singleton
OTUs being omitted. In order to reduce bias caused by variations in sample size, high-quality reads
were randomly subsampled to 923 reads per sample. The alpha diversity was calculated using the
*phyloseq* package R (v.3.5.3). For the analyses of functional genes, Phylogenetic Investigation of
Communities by Reconstruction of Unobserved States (PICRUSt, Langille et al., 2013) was utilized.
The metagenomics prediction table produced from PICRUSt was utilized to produce pathway
abundance profiles using HUMAnN2 (Franzosa et al., 2018). It should be noted that the reconstructed
functional genes were based on the GreenGenes database and not the EzTaxon database used for the
phylogeny.

## 2.3 Physico-chemical Data and Geochemical Analyses

Monthly precipitation for the period in between the cruises (August 2016 to September 2017) were
obtained from the Tropical Rainfall Measuring Mission website (NASA, 2019) in order to gauge the
seasonality (wet or dry; see **Supp. Fig. 1**). The analyses for nutrients encompassing both inorganic (i.e.
Nitrate, $NO_3^-$, Nitrite, $NO_2^-$, Ammonium, $NH_4^+$, Phosphate, $PO_4^-$ and Silicate, $SiO_4^{4-}$) and organic
(dissolved organic nitrate, DON, and dissolved organic phosphate, DOP) fractions were
photometrically determined utilizing a SKALAR San[plus] continuous flow analyser in the State Key
Laboratory for Estuarine and Coastal Research (SKLEC), Shanghai (details described in Sia et al.
2019). $NH_4^+$ and $PO_4^{3-}$ were determined manually following (Grasshoff et al., 1999) while Total
Dissolved Nitrogen, TDN, and Total Dissolved Phopshate, TDP, were determined indirectly by
obtaining the values for $NO_3^-$ and $PO_4^{3-}$ via oxidation with alkaline-persulfate solution (Ebina et al.,
1983). An incubation experiment was set up to study the net primary productivity and respiration rate
of the Rajang River. Triplicates of samples obtained from Belawai (2°13'47.16"N, 111°12'19.04"E)
were incubated in both light and dark set-ups (Refer to **Supp. Table 1** for details).

## 2.4 Statistical Analyses and distLM model

Ordination visualization, non-metric multidimensional scaling (NMDS), and similarity analyses
(ANOSIM) were executed using PRIMER 7 (Clarke and Gorley, 2015) to determine if for example
the various terrestrial source types or different land use determine the structural differences of the
bacterial community. By partitioning the community variation, distance-based linear models





(DistLM) were used to determine the extent of which the bacterial community structure can be
explained by environmental variables (Legendre and Anderson, 1999). Normalizing transformations
of the environmental variables were carried out prior to execution of DistLM analyses. Hellinger
Transformed OTU abundance table was used as the response variable for the variation partition
analysis. The authors would like to note that the distLM models are based on only the August 2016
and March 2017 cruise as there was a lack of physico-chemical data from the September 2017 cruise
due to malfunctioning equipment. Multi-collinearity between variables was tested utilizing the
'Draftsman Plot' function in Primer 7 (Clarke and Gorley, 2006; Supplementary Fig. 1). The authors
would like to note that the distLM models are based on only the August 2016 and March 2017 cruise
as there was a lack of physico-chemical data from the September 2017 cruise due to malfunctioning
equipment. However, it is sufficient to draw linkages between the major drivers of microbial
communities between seasons as Mar 2017 and September 2017 were considered wet seasons based
on the average precipitation (see **Supp. Fig. 1**)


**3.0 Results**
**3.1 Clustering of Samples according to ANOSIM Global Test Scores**
74,690 high quality bacterial sequences were obtained from a total of 117 samples, with 200 to 2,615
sequence reads per sample. The sequences were clustered into 2,087 OTUs at the 97% confidence
interval. Instead of displaying bacterial diversity by station, bacterial communities were grouped
together according to the R scores obtained from the ANOSIM Global test, with the parameters
'cruise', 'source type' and 'land use' showing the highest scores (ANOSIM Global R = 0.737, P <
0.001, **Table 1**).

**3.2 Shifts in bacterial community structure**
The NMDS graph (2D stress score: 0.18, **Fig. 2**), supported ANOSIM results by clustering samples
according to (i) source type and land use as well as (ii) cruises.

The NMDS graph (2D stress score: 0.18, **Fig. 2**) supported ANOSIM results by clustering samples
according to (i) source type and land use as well as (ii) cruises. The X axis (MDS1 scores) clearly
reflects changes in terms of salinity (river-sea continuum) while the Y axis (MDS2 scores) emulates
the different cruises. It is apparent that there were seasonal variations as shown from the lighter shade
points, representing the August 2016 samples, compared to those with darker shades representing both
March 2017 and September 2017 samples (**Fig. 2**). There are apparent overlaps of samples from
mineral soil and brackish peat origin. It can also be observed that there is a gradual shift of samples
from mineral soils and freshwater peat towards brackish and then marine samples, with evident
transitioning between samples.





**3.3 Bacterial Distribution according to source type and cruise**

To further support that the four different source types support distinct bacterial communities, the relative abundance was mapped into a percentage plot (**Fig. 3**).

**Fig 3** show that the phylum *Deinococcus-Thermus* was abundant in freshwater peat and in mineral soils, albeit at a lesser extent compared to freshwater peat. Taking into consideration seasonality, the relative abundance (%) of *Deinococcus-Thermus* drastically decreased in September 2017. Contrary, the abundance of *Cyanobacteria* was greater within marine as well as brackish peat for the cruises of March 2017 and September 2017 but not for August 2016. For the August 2016 cruise, *Cyanobacteria* were found throughout all source types albeit at lower counts compared to the other cruises. Similar changes in bacterial community were observed during different cruises but at different sections of the river. For the marine and brackish peat portions, the cruises of March 2017 and September 2017 were seen to be more similar to each other as compared to the August 2016 cruise with the anomaly of the *Bacteroidetes* phylum. On the other hand, for the freshwater peat and mineral soils, the cruises of August 2016 and March 2017 had greater resemblance towards each other. Furthermore, there was a distinct split in terms of the bacterial community composition for the four source types across all sampling cruises i.e. marine and brackish peat had similar composition and freshwater peat and mineral soils had similar composition. In terms of a river-sea continuum, the most apparent changes in the community composition were observed during March 2017 which presented an almost step-wise change in bacterial community composition.

**3.4 Alpha Diversity Indices**

Based on the Observed indices (**Fig. 4**), mineral soils generally had the highest counts of unique OTUs. However, during the September 2017 cruise, the freshwater region had the highest values. Based on the Chao1 indices, there was a significant effect of the source type on the observed richness ($p<0.001$), with increasing values from marine to mineral soils. In the March 2017 and September 2017 cruise, the Chao1 indices were found to have greater variability as compared to the August 2017 cruise. For the September 17 cruise, the values for Chao1 across the brackish peat, freshwater peat as well as mineral soils were all observed to have increased values of Chao1. According to the Shannon indices, the diversity of the microbial communities were significantly different along the different source types ($p<0.001$). In the dry season the Shannon indices were found to be higher than that found in March 17 and September 2017 samples, except for the Brackish peat September 2017 samples. In terms of the Simpson diversity indices, the August 2016 season was found to have the higher values as compared to the March 2017 and September 2017 season.





Based on the effects of land use on the diversity indices (**Fig. 5**), the sites which are surrounded by human settlements had higher observed indices (regardless of the cruise), with the exception of the Shannon indices in August 2016. Samples surrounded by secondary forest had the second-highest values with samples from August 2016 repeatedly higher than the other two cruises. There were significant differences (p<0.001) between samples from the coastal region with generally lower indices compared to upstream samples.

### 3.5 Functional Profile of Bacterial Communities

Based on the KEGG pathways (**Fig. 6**), the functional profiles of the microbial communities were predicted for the Aug 2016 and Mar 2017 samples. The metabolic pathways that were selected were based on the active pathways that were exhibited, including the metabolism of Nitrogen, Carbohydrate, Methane and Sulfur metabolism. The main functions found were oxidative phosphorylation (20.09%), carbon fixation pathways in prokaryotes (19.00%) and methane metabolism (18.36%), respectively. This was then followed by nitrogen metabolism (11.50%), carbon fixation in photosynthetic organisms (7.67%), inorganic ion transport and metabolism (5.68%). The remaining functional groups were photosynthesis, sulphur metabolism, inositol phosphate metabolism, phosphotransferase system (PTS), carbohydrate metabolism, phosphonate and phosphinate metabolism and lastly mineral absorption (4.92%, 4.31%, 2.96%, 2.34%, 1.83%, 1.11% and 0.23%, respectively). From **Fig. 6**, it can be seen that the functional gene profiles that were derived from the metagenomic profile were very similar. This was similar to a study by Fortunato and Crump (2015) who observed that the average similarities of the functional gene profiles were 82% from river to ocean. In terms of gene abundances, the March 2017 samples (wet season) were found to have higher gene abundances with the highest counts in brackish peat followed by marine samples. However, marine samples in August 2016 displayed slightly higher gene counts compared to the brackish peat.

### 3.6 Distance-based Linear Model of bacterial communities and environmental parameters

Marginal DistLM was performed in order to gauge the extent of physicochemical parameters or environmental variables accounting for a compelling proportion of variation in the bacterial communities. Salinity was the single best predictor variable explaining bacterial community variation (15.27%), followed by Dissolved Inorganic Phosphate at 10.57%. The remaining physico-chemical parameters were dissolved oxygen (9.64%) and Suspended Particulate Matter (6.55%) whereas for the biogeochemical parameters, Silicate (9.27%), Dissolved Organic Phosphate (8.04%), Dissolved Organic Nitrogen (6.37%), Dissolved Organic Carbon (5.27%) and lastly Dissolved Inorganic Nitrogen (4.29%, respectively) made up the remaining variables (all variables P = 0.001, except for DIN, P=0.002).



Significant vectors of environmental variables ($R^2$>0.3892, P <0.001) were calculated based on a
linear model (DistLM) and plotted against the bacterial community composition as shown in **Fig 7**.

From **Fig. 7**, the distLM model clustered samples from the August 2016 cruise away from the samples
of the March 2017 cruise (as seen from the plot points with lighter shades as August 2016 and darker
shades as March 2017).  Samples originating from the brackish peat as well as marine region (August
2016) irrespective of land use were shown to cluster more strongly towards salinity (as shown from
the longer vector from salinity) as well as DIN and DOP, followed by DIP. On the other hand, the
brackish peat as well as marine samples from the March 2017 were found to cluster in between DIP
and DO. In addition, the samples from August 2016 for freshwater peat and mineral soil -irrespective
of land use- clustered towards silicate and DON whereas for March 2017, the samples were shown to
cluster towards the SPM vector. Lastly, it was found that samples which are of peat origin were also
adjacent to the DOC vector.


**4.0 Discussion**
This study presents seasonal and spatial distribution of particulate-attached and free-living bacteria in
the longest river in Malaysia in an attempt to map the bacterial community composition of the water
column across several habitats with relation to the riparian zones and anthropogenic activities in a
river-to-sea continuum. Our dataset allows comparison of the microbial community across two
dimensions: spatial biogeography from headwaters to the coastal zone as well as through time
(seasonally). The rich supporting dataset also allows us to assess underlying nutrient dynamics
influencing the microbial communities.

**4.1 General bacterial community composition**
The core microbial communities along the Rajang River-South China Sea continuum consist of
*Proteobacteria*, *Firmicutes*, *Actinobacteria*, *Bacteroidetes*, *Deinococcus-Thermus* and *Cyanobacteria*
in varying abundances (**Fig. 3, Supp. Fig. 5**) indicate high variation within the system. Staley et al.
(2015) proposed that variability in microbial communities were less due to the presence/absence but
likely due to shifts in relative abundance of OTUs. As shown in **Fig. 3** and **Supp. Fig .5**, the bulk
bacterial taxa were restricted to a relatively small number of assemblages. However, due to the
heterogeneity of the Rajang River, substantial shifts in OTU diversity were shown, while exhibiting
successional changes in community composition downstream, there were abrupt shifts in terms of
richness and diversity as well as bacterial distribution which was structured according to macro-scale
source types. While there were shifts in the community composition, based on the OTU overlaps,
particle association of the samples were not apparent (**Supp. Fig. 3, Supp. Fig. 9**). The similar
bacterial community structure in terms of particle association was in line with studies by Noble et al.,



(1997) and Hollibough et al., (2000) in the Chesapeake Bay (winter season) and San Francisco Bay,
respectively. Hollibough et al., (2000) further supported that the difference or similarity of the particle
association of bacterial community was due to the origin as well as composition of the particles,
particularly in marine snow or estuarine particles. In the aforementioned study, there was limited
metabolic divergence and similar communities between the estuarine turbidity maxima and the river
samples. Due to the short residence time, the rapid exchange of organisms likely reduced the
divergence of phylogenetic composition. The short residence time in the Rajang River as reported by
Müller-Dum et al. (2019) likely reflected similar a similar scenario with the San Francisco Bay.

**4.2 Diversity and shifts in bacterial communities along the Rajang river-South China Sea continuum**

When comparing with other rivers, the predominance of the *Proteobacteria* phylum, especially within
the brackish peat region (**Fig. 3, Supp Fig. 5**) was similar to a recent study on the Pearl River Delta
(Chen et al., 2019). In another study by Doherty et al. (2017) on the mainstem of the Amazon River (a
blackwater influenced river, similar to the Rajang River), *Actinobacteria* were much more abundant
(25.8%) compared to the Rajang River (11.95%). However, the second-most abundant taxon was the
*Proteobacteria* (*β-Proteobacteria*) which peaked during seasons of high discharge. The same pattern
of peaking during high discharge can be observed in the Rajang River with considerably higher
relative abundance in the wet season (**Fig. 3**). This could be a result of the intense rainfall that led to
the large input of freshwater (Silveira et al., 2011), and ultimately resulting in a "trickling" over
microbial pattern from the freshwater to the brackish region. The predominance of *β-Proteobacteria*
in the freshwater region and the predominance of α- and γ-*Proteobacteria* (**Supp. Fig. 4**) in the
estuarine region is typical as the main group in seawaters (Nogales et al., 2011)and similar to findings
by Silveira et al. (2011) on the bacterioplankton community along the river-to-ocean continuum from
the Parnaioca River towards the Atlantic Ocean. Hence, this shows that salinity exhibited a strong
influence on the abundances of *Proteobacteria* and *Firmicutes*.

Among the proteobacterial classes, γ-*Proteobacteria* was the most dominant, followed by α-
*Proteobacteria*. The high abundance of γ-*Proteobacteria* is in line with Fuchsman et al. (2012) which
states that the group is commonly regarded as particle-associated bacteria. When compared across the
river-to-sea continuum, the low abundance of β-*Proteobacteria* is in contrast to other literature
(Brown et al., 2015; Ghai et al., 2012) whereby the majority of freshwater systems has β-
*Proteobacteria* as the most dominant taxa, as the determination takes into account the estuarine as
well as the marine regions. The phylum *Proteobacteria* was dominant in all the samples, indicating its
role in nitrogen cycling (Yang et al., 2013). The presence of *Protebacteria* in its role in nitrogen
cycling is complementary to the *Cyanobacteria* blooms which occur as evidently shown in **Fig. 3**.
Furthermore, the higher presence of *Chloroflexi* (Ward et al., 2018) and *Cyanobacteria* (Guida et al.,
2017) within the marine and brackish peat region indicated its probable role in carbon fixation as
reflected by the higher gene counts (carbon fixation pathways in prokaryotes) in the marine and
brackish peat regions as compared to the freshwater peat and mineral soil (**Fig .6**). Furthermore, the
presence of the genus *Sphingomonas* indicated the presence of purple-sulfur bacteria which were able
to utilize carbon dioxide (carbon fixation pathways in prokaryotes) and oxidation of Hydrogen Sulfide
(sulphur metabolism) (Pfennig, 1975) (**Fig. 6**). In the case of *Firmicutes*, the higher abundance of
*Firmicutes* in the brackish region was reflective of the overall production as opposed to selective
growth of the particular source type, as *Firmicutes* were found throughout all four source types. The
highest presence of *Deinococcus-Thermus* (**Fig. 3**) was found in freshwater peat environments,
indicating its preference for the aforementioned environment. This is interesting to note as most
studies on bacterial community composition show that the phylum *Deinococcus-Thermus* occurs in a
higher abundance in extreme environments such as in hot springs (Zhang et al., 2018b) or in studies
that are analogous for Mars (Joseph et al., 2019). In contrast, most extreme environments show that
*Deinococcus-Thermus* was found in low percentages such as in Antarctic marine environments (1%,
Giudice and Azzaro, 2019), 1.5% in hypersaline soils (Vera-Gargallo et al., 2019) as compared to the
Rajang River. When taking into consideration the major genera, there is a fundamental shift in
bacterial community composition along the continuum (**Fig 3**, **Fig. 4**) together with the bacterial
richness and diversity indices, there was a distinct difference between the dry season (August 2016)
and both wet seasons, with September 2017 having higher observed indices while the March 2017,
while being a wet season as well had lower or variable observed indices. This difference in the two
wet seasons could be the due to the different stages of phytoplankton bloom as mentioned earlier
whereby the September 2017 was during an algal bloom while the March 2017 was after an algal
bloom event. This was reflected in the Simpson index as well as the indices for September 2017 being
lower than those of the August 2016 or March 2017 samples. Similarly, Zhou et al. (2018)
demonstrated that the Simpson Indices for bacteria increased after the onset of an algal bloom
(Brackish peat, September 2017) whereas the Shannon indices was at the lowest (Brackish peat,
March 2017) (when assuming that the region in which phytoplankton blooms occur is the brackish
peat region). Overall, there was greater diversity (based on Shannon Indices) in the dry season
(August 2016) than the wet seasons (March and September 2017) whereas there were greater OTUs in
the wet season (Observed index). The decrease in richness and evenness was similar to a study
conducted by Savio et al. (2015)in which the bacterial evenness and richness declined downriver,
which is in line with the River Continuum Concept (Vannote et al., 1980). The presence of peat did
not affect the alpha-diversity indices which is reflected in the shift in taxa occurring from freshwater
(which includes freshwater peat) towards the saline region (which includes brackish peat). Dominant
phyla typically found in Malaysian peat swamps such as *Proteobacteria* (Kanokratana et al., 2011;
Too et al., 2018; Tripathi et al., 2016) are found throughout the Rajang river whereas *Acidobacteria* is
not a major phylum in the Rajang river.




### 4.3 Factors determining bacterial community composition

While there is difficulty in assessing microbial communities in lotic environments due to the
heterogeneity of the physicochemical parameters that lotic environments are subjected to (Zeglin,
2015), the major drivers of microbial communities should still be assessed. While only two cruises
(August 2016 and March 2017) were used due to the lack of physico-chemical data for the September
2017 cruise, it is sufficient to draw linkages between the major drivers of microbial communities
between seasons as March 2017 and September 2017 were both considered wet seasons based on the
average precipitation (see **Supp. Fig. 1**). As shown in **Fig. 2**, it can be observed that there is a
continual shift in microbial communities, suggesting mixing of the microbial communities from the
headwaters to the coast (Fortunato et al., 2012) which has also been observed along the Upper
Mississippi River (Staley et al., 2015) and along the Danube River (Savio et al., 2015). Based on the
linear model (**Fig. 7**), salinity is an important factor in driving the shift in microbial communities
(**Table 2**), akin to findings by Herlemann et al. (2011) along a 200 km salinity gradient in the Baltic
Sea**.** The dispersal of taxa of microbial communities from fresh to marine waters faces a strong barrier
due to salinity (Fortunato and Crump, 2015), likely explaining the reduced relative abundances of
*Chloroflexi* upstream and in turn the reduced *Deinococcus-Thermus* downstream (**Fig. 3**). Such
dispersals are further influenced by transitional waters such as estuaries and plumes whereby the
microbial communities are exposed to rapidly changing physico-chemical conditions such as salinity
gradients, nutrients, temperature as well as sporadic anthropogenic inputs (Crump et al., 2004). While
the distribution of the core microbial communities are indicative of the river-sea continuum, it is
noteworthy that several phyla were distinctly associated with specific source types. The distinct shift
in bacterial taxa for example from Freshwater to Brackish waters (and lack thereof between
freshwater peat and brackish peat; **Fig. 3**) indicates that peat did not have a significant effect on the
distribution of bacterial taxa. This is further supported by the fact that DOC (as a proxy for organic
matter of peat origin) only accounts for 5.27% of the community variation (**Table 2**). A study on
blackwater rivers in the Orinoco Basin, Venezuela (Castillo et al., 2004) showed that increased DOC
resulted in higher bacterial production, however, the change in bacterial production is not a reflection
of its influence on the community composition. This was supported based on a simple respiration
experiment conducted in Aug 2016 (**Supp. Table 1**) whereby the respiration rate ($0.44 \pm 016$ g DO L$^{-1}$
d$^{-1}$) was higher than that of the primary production rate ($0.39 \pm 0.08$ g DO L$^{-1}$ d$^{-1}$).

According to Peter et al. (2011) and Wilhelm et al. (2015)salinity, DIP (biogeochemical parameter)
and Dissolved Oxygen (physical parameter) had major impacts on the distribution of species. This is
neatly supported by the distribution of samples on the distLM fitted dbRDA graph (**Fig. 7**) whereby
the affinity for each of the samples correlates to the physical environment (e.g. the samples which
group along the salinity vector were the samples which correlate with the marine as well as brackish



peat region. Samples influenced by dissolved oxygen (**Fig. 7**) are from the estuarine region which showed an almost anoxic zone (refer to **Supp. Fig. 7**). The low availability of oxygen is mirrored in higher counts (samples belonging to the brackish peat category showed highest counts regardless of phyla as well as season; **Supp. Fig. 5**). Higher counts (particularly *Chloroflexi* and *Cyanobacteria*) do, however, not reflect higher primary production within this zone. While zones of coastal estuaries are usually deemed to have higher primary productivity, it can be inferred that the depletion in oxygen and higher $pCO_2$ emissions (Müller-Dum et al., 2019) within the brackish peat region of the August 2016 campaign was a result of high bacterial productivity. This can be further supported by the high suspended particulate matter (SPM) as a proxy of turbidity of the brackish peat (**Supp. Fig. 7**) which may have resulted in the reduced primary productivity, which in turn can explain the lower dissolved oxygen values. As aforementioned earlier, the respiration rate ($0.44 \pm 016$ g DO L$^{-1}$ d$^{-1}$) was higher than that of the primary production rate ($0.39 \pm 0.08$ DO L$^{-1}$ d$^{-1}$). This was similar to a study in the Scheldt River whereby the higher bacterial production occurred in the turbidity maxima together with the depletion of oxygen (Goosen et al., 1995). However, the relative abundance of bacterial OTUs were higher in the estuary as well as marine region, reflecting that while the microbial communities are structured by salinity, the abundance is more a reflection of the nutrients available, especially in estuaries which exhibit circulation patterns which can result in localised nutrient-rich conditions (They et al., 2019). This was supported by the higher relative abundance of oxidative phosphorylation genes as well as nitrogen metabolism within the brackish peat and further supported by Jiang et al. (2019) demonstrated through incubations studies whereby N transformations in the Rajang River estuary mixing zone was higher than in the Rajang River and coastal region.

While the development of unique community structures is strongly influenced by spatial factors, an influence of seasonality could also be observed with samples from March 2017 being distinctly different from the other two cruises (August 2016 and September 2017; **Supp. Fig. 3**). Seasonal variability was also observed between the source types, particle association and down to the genus level (**Fig. 2**, **Supp. Fig. 3** and **Supp. Fig. 6**). Based on the precipitation as an indicator of the seasonality, a probable "transitioning" phase was observed in the dry season (August 2016) with the microbial communities being more alike with the March 2017 samples (Fig. 8) when comparing both wet seasons (March 2017 and September 2017). Within the phylum rank **(Fig. 3)**, the presence of *Cyanobacteria* during the March and September 2017 cruises indicates the influence of seasonality. However, while March 2017 and September 2017 were both considered to be wet seasons based on the precipitation, in terms of the relative abundance, there are considerable differences between the two cruises. The greater abundance of *Bacteroidetes* in March 2017 may be indicative of the community composition adjusting following an algal bloom (Pinhassi et al., 2004). In the September 2017 season, it is probable that the time sampled was still during an algal bloom, as indicated by the higher abundance of *Cyanobacteria*. Moreover, the shifts in community composition from Aug 2016





to March 2017 and from March 2017 to September 2017 are indicative of the influence of seasonality.
While March 2017 and September 2017 were similar in terms of seasons, September 2017 had higher
precipitation during that month, which led to higher run-off from the riparian region as compared with
the March 2017 wet season. This could have led to the increase in cyanobacteria, which was also
reflected increase of picoplankton size class during the wet season where it is hypothesized that the
September 2017 might be more optimal for picoplankton proliferation (**Supp. Fig. 8**). Furthermore,
in comparison, August 2016 and March 2017 were similar in terms of the proportion of the relative
abundance of the community composition (**Fig. 3**).

**4.4 Possible pathogenic bacteria and/or anthropogenic influence and land-use change**
According to Reza et al. (2018) the taxa *Flavobacterium* is a potential fish pathogen which is
commonly found in freshwater habitats (Lee and Eom, 2017) as well as coastal pelagic zones (Eilers
et al., 2001). In the Rajang river, it is the sixth most abundant class (**Supp. Fig. 5**). This is cause for
concern as it was found to be high in the coastal regions as well as brackish regions where fisheries
and fishing activities are concentrated. Furthermore, the *Cytophaga-Flavobacterium-Bacteroidetes*
group, or rather known as the CFB group, are commonly associated with humans (Weller et al.,
2000), reflecting anthropogenic influences on the samples, especially within the brackish areas which
has several human settlements and plantations. Lee-Cruz et al. (2013) demonstrated that conversions
of oil palm plantations from tropical forests are much more severe as compared to logged over forests
in terms of bacterial community composition whereby logged over forests was shown to exhibit some
resilience and resistance (to a certain extent). There has been little to no literature regarding the
changes in microbial community composition as a result of land-use changes that occur within this
region, particularly throughout the catchment area of the Rajang River. However, the results obtained
from this study evidently suggest that the run-off from anthropogenic activities alters the microbial
community composition. Anthropogenic disturbances, in particular settlements and logging
(secondary forest), led to higher diversity indices (**Fig .6**). On the contrary, sites surrounded by oil
palm plantations displayed the lowest diversity indices, supporting results by Mishra et al. (2014) who
found similar results in peatlands. Furthermore, the OTU overlapping of major anthropogenic
activities (i.e settlements and oil palm plantations) in **Supp. Fig. 10** reflected the possibility of higher
abundance of generalists as compared to sensitive species (Jordaan et al., 2019) as microbial
communities generally adapt to permanent stress events such as increased concentrations of inorganic
or organic nutrients. In another study conducted by Fernandes et al. (2014), anthropogenically-
influenced mangroves had 2x higher the amount of *γ-Proteobacteria* compared to pristine mangroves.
This was similar to the March 2017 cruise along the Rajang River, whereby *γ-Proteobacteria* was the
predominant class in the marine and brackish peat region along with the significant increase in
*Bacteroidetes* as aforementioned, which can be associated to anthropogenic activities. On the other
hand, during the dry season, the diversity of the "less-disturbed" region was higher than the disturbed





regions. However, it should be noted that the coastal zone generally has the lowest richness and
diversity amongst the other regions regardless of the presence or absence of anthropogenic activities.
Hence, the extent of salinity intrusion may also result in the loss of diversity and richness of the
microbial communities (Shen et al., 2018) in the Rajang River.

**5.0 Conclusion**
This study represents the first assessment of the microbial communities of the Rajang River, the
longest river in Malaysia, expanding our knowledge of microbial ecology in tropical regions. The
predominant taxa are *Proteobacteria* (50.29%), followed by *Firmicutes* (22.35%) and *Actinobacteria*
(11.95%). The microbial communities were found to change according to the source type whereby
distinct patterns were observed as a result of the changes in salinity along with variation of other
biogeochemical parameters. Alpha diversity indices indicate that the microbial diversity was higher
upstream as compared to the marine and estuarine regions whereas anthropogenic perturbations led to
increased richness but less diversity in the less pristine environments compared to those that were
more pristine. Even though there were observed changes in bacterial community composition and
diversity that occur along the Rajang River to sea continuum, the PICRUST predictions showed minor
variations. Areas surrounded by oil palm plantations showed the lowest diversity and other signs of
anthropogenic impacts included the presence of CFB-groups as well as probable algal blooms. In
order to further gauge and substantiate the functional and metabolic capacity of the microbial
communities within each specific source type, metaproteomics as well as metabolomics should be
carried out along with mixing experiments in order to further gauge the response of the microbial
communities towards anthropogenic perturbations as well as the role of microbial communities in
degrading peat-related run-off from the surrounding riparian regions.

**6.0 Acknowledgements**
The authors would like to thank the Sarawak Forestry Department and Sarawak Biodiversity Centre
for permission to conduct collaborative research in Sarawak waters under permit numbers
NPW.907.4.4(Jld.14)-161, Park Permit No WL83/2017, and SBC-RA-0097-MM. Special mention to
the boatmen who helped us to collect samples, in particular Lukas Chin and his crew during the
Rajang River cruises. Also, the authors are very grateful to Dr. Kim Mincheol of KOPRI for
providing the mothur codes and supercomputer for processing the sequences. We would also like to
thank Patrick Martin for providing DOC measurements and Denise Müller-Dum for providing SPM
measurements. Gonzalo Carassco, Nagur Cherukuru as well as student helpers from UNIMAS,
Swinburne Sarawak, SKLEC and NOCS greatly aided with the logistics and fieldwork. M.M.



acknowledges funding through Newton-Ungku Omar Fund (NE/P020283/1), MOHE FRGS 15 Grant
(FRGS/1/2015/WAB08/SWIN/02/1) and SKLEC Open Research Fund (SKLEC-KF201610).




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



**Tables**

**Table 1:** ANOSIM Global Test scores based on various parameters

| Parameters tested, 999 permutations, random sampling | ANOSIM Global Test, R | P value |
|---|---|---|
| Cruise (Wet/Dry season) | 0.439 | 0.001 |
| Source Type | 0.422 | 0.001 |
| Land use | 0.182 | 0.001 |
| Particle Association | 0.037 | 0.001 |
| Source Type, Land use | 0.415 | 0.001 |
| Cruise, Source Type, Particle Association, | 0.708 | 0.001 |
| Cruise, Source Type, Land use | 0.737 | 0.001 |


**Table 2:** Proportion of combined community variation based on marginal DistLM test that is
explained by each predictor variable using two cruises (August and March 2017)

| Category | Variable | Pseudo-F | *P*-value | Proportion explained (%) |
|---|---|---|---|---|
| Physico-chemical parameters | Salinity | 9.6128 | 0.001 | 13.42 |
| | Dissolved oxygen | 6.6151 | 0.001 | 9.64 |
| | SPM | 4.3486 | 0.001 | 6.55 |
| Biogeochemical parameters | DIP | 4.2218 | 0.001 | 10.57 |
| | Silicate | 9.269 | 0.001 | 9.27 |
| | DOP | 5.4246 | 0.001 | 8.04 |
| | DON | 4.2218 | 0.001 | 6.37 |




**Figure Captions**

**Fig. 1**: Location of Rajang River within Sarawak, Malaysia (inset). (A) shows the stations sampled during three (3) different cruises; August 2016 (red triangles), March 2017 (blue circles) and September 2017 (cyan diamonds). (B) GIS data from 2010 (Sarawak Geoportal, 2018) indicating various forest types. Red colour represents non-forest areas (2010), yellow represents non-forest areas (2013), light green represents primary forests, teal represents secondary forests whereas dark green represents potential peat swamp forests.

**Fig. 2**: Non-metric Multi-dimensional Scaling (NMDS) graph of samples according to cruise, source type as well as land use.

**Fig. 3:** Relative abundance (%) of dominant bacterial (at phylum level, top 10) along the various source types (Marine, Brackish Peat, Freshwater Peat, Mineral Soils) across 3 cruises/seasons

**Fig. 4**: The calculated α-diversity indices (Observed, Chao1, Shannon, Simpson and Inverse Simpson) of the four different source type along the salinity gradient.

**Fig. 5**: The calculated α-diversity indices (Observed, Chao1, Shannon, Simpson and Inverse Simpson) of the Land Use types (Coastal Zone, Coastal Zone with Plantation (OP) influence) Coastal Zone with Plantation (Sago and Oil Palm influence), Human Settlement, Oil Palm and Sago mixed Plantation, Oil Palm Plantation and Secondary Forest)

**Fig. 6**: The relative abundance of predicted functional profiles in the four source types across two seasons based on KEGG Pathways

**Fig. 7**: Distance-based Redundancy Analysis (dbRDA) plot based on a linear model (DistLM) and plotted against the bacterial community composition.





**Figures**

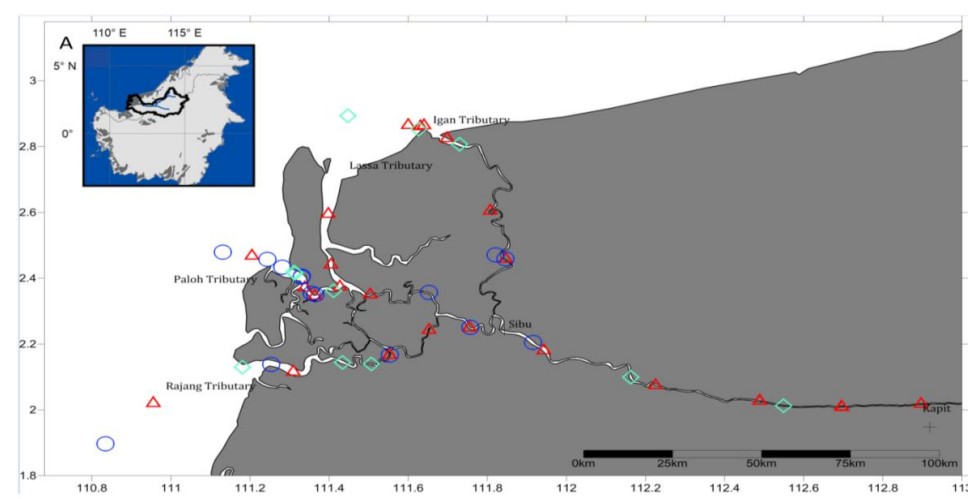



**Fig. 1**





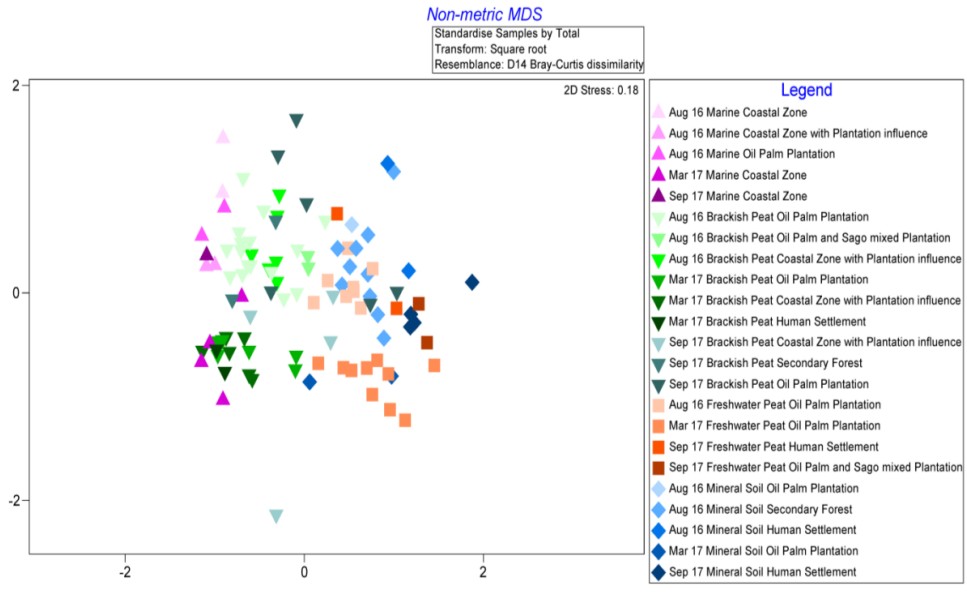

**Fig. 2**

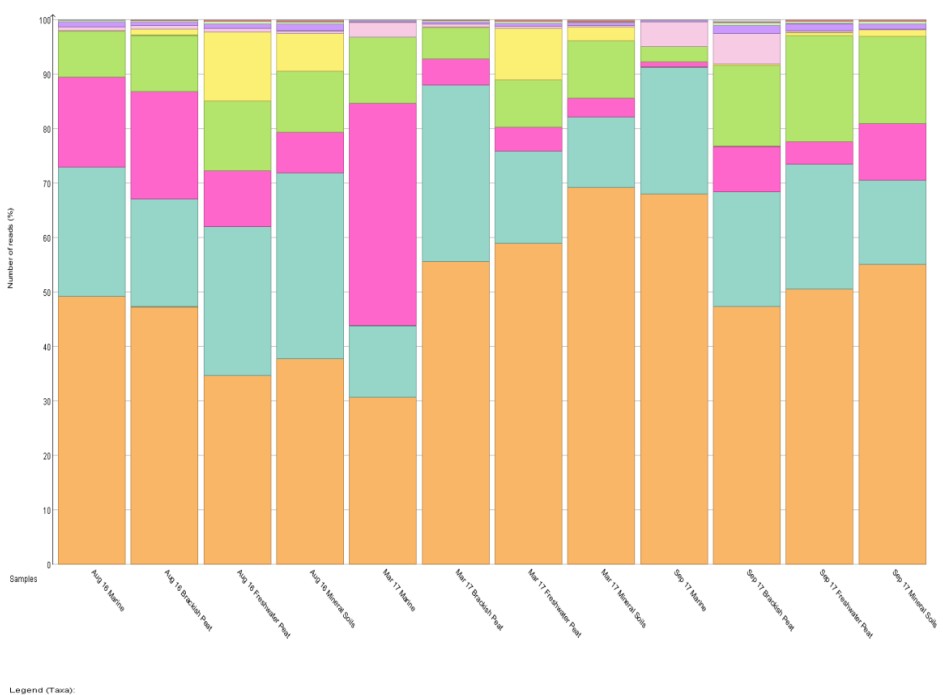



**Fig. 3**





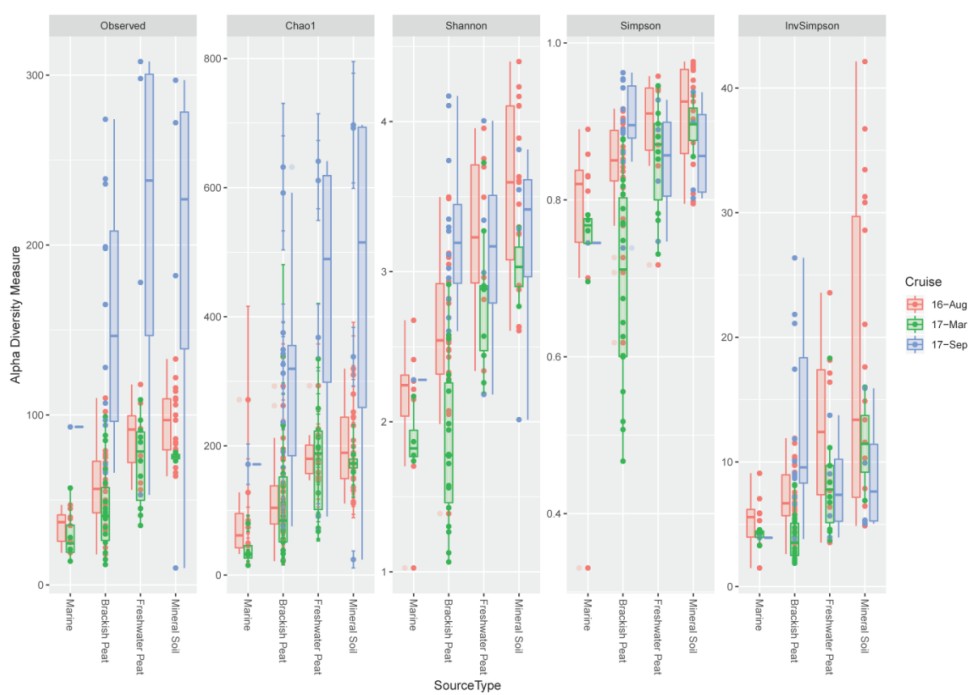


**Fig. 4**


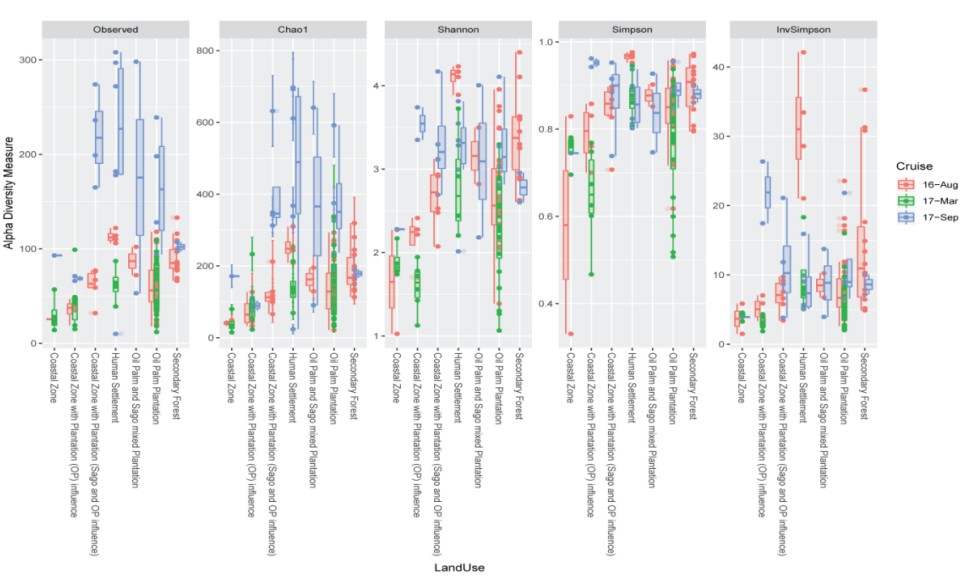


**Fig. 5**



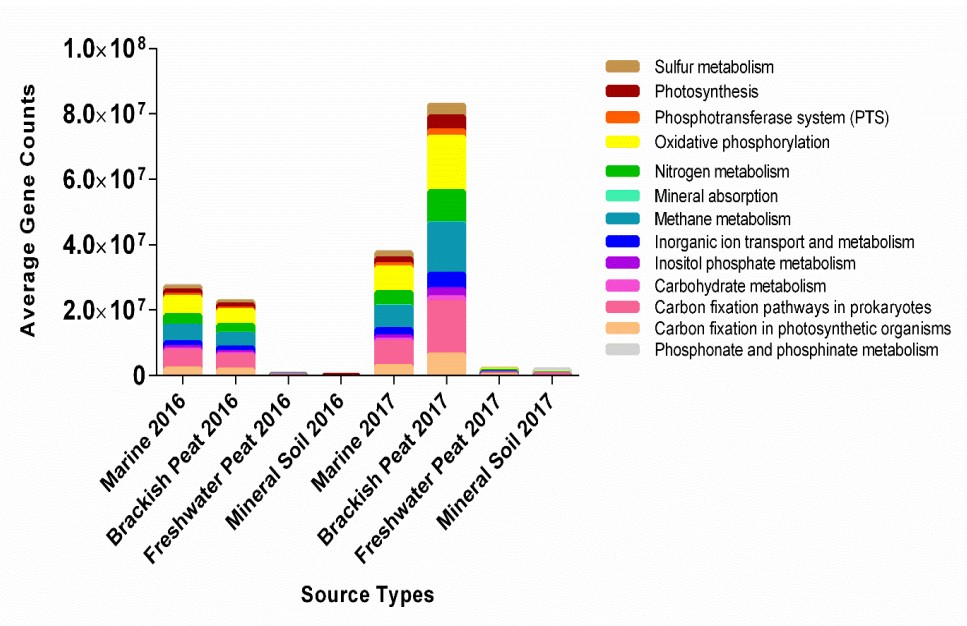

**Fig. 6**

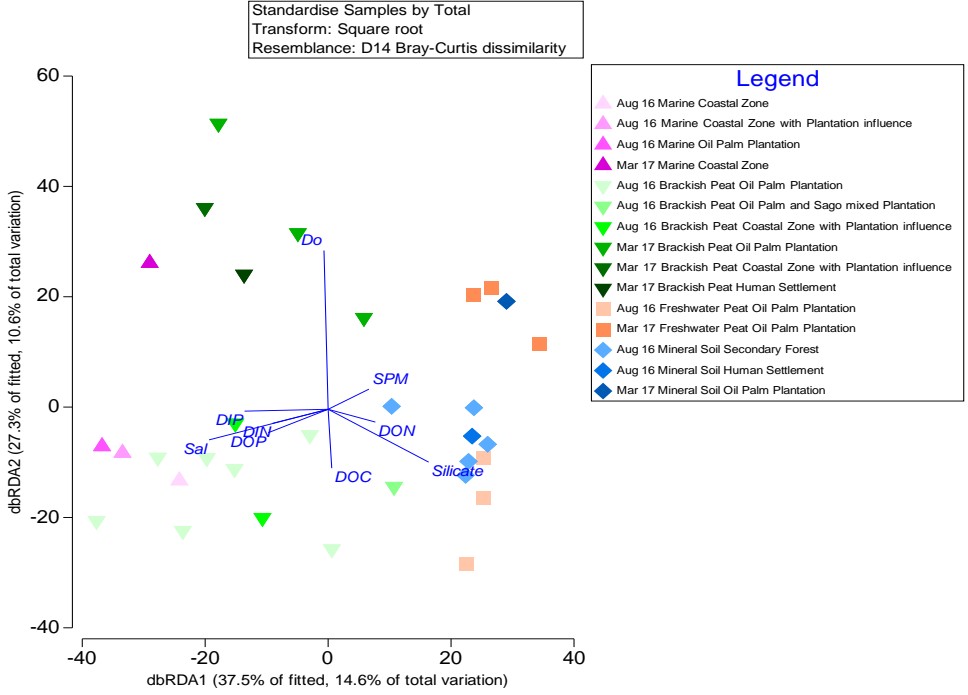


**Fig. 7**