# Peer review of "Biogeographical distribution of Microbial Communities along the Rajang River-South China Sea Continuum"

_Biogeosciences, 2019_

## Referee Comment (RC1) · Anonymous Referee #1 · 15 Jul 2019

Manuscript: bg-2019-214 Title: Biogeographical distribution of Microbial Communities along the Rajang River-South China Sea Continuum

A. General comments This study addresses the underlying factors that may influence the spatial and seasonal distribution of the prokaryotic communities and nutrient dynamics along the Rajang River, South China. Although the results of this study are valid and interesting, there are several points that need to be addressed. 1) Dividing sampling cruises into "wet season" and "dry season" may be more beneficial than referring to them individually. Authors mentioned both wet and dry seasons in the Methodology section (section 2.1), however, sampling cruises associated with each are lacking.

2) The site map (Figure 1A) currently shows sampling points throughout the river with source types (Figure 1B), but illustration of anthropogenic activities along the river is missing. It will be helpful to add these as it's not clear which sections of the river are impacted by which activities. 3) Additional statistical analysis, such as PERMANOVA, may be used to infer the impact of anthropogenic activities (e.g. human settlements, effluents, transportation and sand dredging) and source types on beta diversity. Much of the Results and Discussion sections revolve around alpha diversity indices but very little is mentioned about beta diversity. 4) Potential functionality inferred from PICRUST showed clear distinction between samples when comparing source type. It would be interesting to see if potential functionality differed too such an extent when samples are compared by anthropogenic perturbations. 5) The general flow of the Discussion section needs improvement. The significance and contribution of the study will have a bigger impact when the Discussion is presented clear and logically. Also, the authors should double check the tense (present/past/passive) for each section. 6) Recheck format of in-text references. Not all citations are written in the same format.

B. Specific comments 1.0 Introduction p. 4 paragraph 6: The authors aimed to study microbial diversity and potential function in the Rajang River. Although this study is the first to investigate microbial diversity along a freshwater-marine gradient, with a tropical peatland component, the importance of the river in Malaysia and clear objectives need to stated.

2.0 Methodology 2.1 Study area and sampling strategy p. 5 line 136-139: "According to Wetlands International (2015), the land surrounding the study sites is characterized by a range of anthropogenic activities, ranging from oil palm and sago plantations to human settlements as well as transportation and sand dredging activities (Fig. 1(B))." This is not clear from Fig 1B. Colours are associated with forested or non-forested lands, however, the map does not depict the different anthropogenic activities along the river.

2.2 Pyrosequencing and Bioinformatics Analyses • Change "Pyrosequencing"

to "Illumina sequencing" in the subheading. The authors did not perform 454-pyrosequencing but Illumina sequencing • Did ACE also extract DNA from samples? If that's the case, the authors should mention this in the beginning of this section.

2.4 Statistical Analyses and distLM model • The authors used db-RDA to determine the impact of environmental variables on microbial diversity. The same method can be used to determine which parameters have an influence on specific bacterial taxa. Likewise, Spearman/Pearson correlations can be drawn between environmental parameters and taxa. The information inferred from these additional analyses can help the authors to link certain taxa to specific source types or pollution sources. • I also suggest that anthropogenic inputs should be divided into the following categories: human settlements, effluents (from both palm oil and sago plantations), transportation and sand dredging. Variation partitioning, if possible, may then be used to determine which anthropogenic input, or source type, had the biggest impact on bacterial diversity along the river.

3.0 Results 3.3 Bacterial Distribution according to source type and cruise This section may be improved by organizing it into the following paragraphs: • Mention the dominant taxa and their relative abundances. The author mentioned this in the Discussion section (section 4.1 line 333-335) but not in the Results section. • Which taxa (dominant or specialized) were more predominant at specific source types and/or seasons? • In this section, the authors acknowledge a higher Cyanobacterial abundance for the September 2017 marine and brackish peat samples. In the Discussion section they refer to the higher Cyanobacterial counts as "blooms" without prior testing (e.g. chlorophyll-a) as a proxy measure to confirm algal blooms. Since Cyanobacteria are more pronounced during warmer months, and anthropogenic activities close to the sampling areas can cause nutrient input and thus proliferation of Cyanobacteria, how valid is the assumption/statement made in the manuscript without additional measurements?

3.4 Alpha Diversity Indices • How did the authors calculate the effect of land use
and source type on alpha diversity indices? This was not mentioned in the Methodology section. • Instead of comparing indices between cruises, the authors can make comparisons between seasons (e.g. compare the entire wet season with the dry season).

4.0 Discussion • I recommend that the authors re-write certain paragraphs of the Discussion section so that it may have a bigger impact on readers. • Instead of naming all the different types of taxa in the river, focus on the important ones and what their roles are. How does the environment and different inputs (source types and anthropogenic activities) impact these taxa?

4.4 Possible pathogenic bacteria and/or anthropogenic influence and land-use change • Was Flavobacterium the only potential pathogen identified? • I would suggest to start the paragraph with anthropogenic influence and land-use change. A second, shorter paragraph can discuss the potential pathogens

C. Technical comments 1.0 Introduction p. 3: Combine paragraphs 2 and 5. Both are discussing lotic environments and nutrient cycling; it will thus make more sense to combine these two. p. 4 line 93-93: Due to their high diversity and fast generation time, microbial communities are the first responders to environmental changes…... p. 4 line 96: Liao et al. (2019) showed that p. 4 line 97: delete "further" p. 4 line 97: Bruland et al. (2008) demonstrated that the p. 4 line 99-102: "Thus, as the Rajang River experiences two monsoonal seasons (Sa'adi et al., 2017) and is subjected to anthropogenic disturbances (Gaveau et al., 2016; Miettinen et al., 2016), it is thus fundamental to take into consideration both seasonal and anthropogenic influences on the microbial communities of the Rajang River." This forms part of the aim and objective and should rather move to last paragraph p. 4 line 115: delete "hypothesized" p. 4 line 120-121: as well as anthropogenic disturbances (e.g. human settlements and plantations) on microbial succession. p. 4 line 121-122: Delete "Linear models are used to examine the relationship between the microbial community structure and their environment."

[Figure]

2.0 Methodology 2.1 Study area and sampling strategy p. 5 line 130: The region... p. 5 line 134: small tributaries p. 5 line 142: Which months were associated with the wet and dry season, respectively? p. 5 line 149: Approximately 250 – 500 mL of water.... p. 5 line 153-156: A total of 117 filters were recovered (1 x 3.0 $\mu$m filter was discarded due to contamination) and immediately stored at - 20°C.

2.2 Pyrosequencing and Bioinformatics Analyses p. 5 line 160: Briefly, fastq files generated... p. 6 line 161: quality trimmed with fastqc, primer sequeces.... p. 6 line 162-163: High quality sequences were subsequently processed using the Mothur pipeline. p. 6 line 164: SILVA database p. 6 line 171: potential functional genes

2.3 Physico-chemical Data and Geochemical Analyses p. 6 line 179: in-between the cruises p. 6 line 189-191: Belawai samples (2°13'47.16"N, 111°12'19.04"E) were used in an incubation experiment to study the net primary productivity and respiration rate of the Rajang River. Technical triplicates were incubated in both light and dark set-ups (Refer to Supp. Table 1 for details).

2.4 Statistical Analyses and distLM model p. 6 line 195-197: to determine if the various terrestrial source types or different land use impacted bacterial community composition. p. 7 line 199: what type of normalization method was used? p. 7 line 202-204: "The authors would like to note that the distLM models are based on only the August 2016 and March 2017 cruise as there was a lack of physico-chemical data from the September 2017 cruise due to malfunctioning equipment." Delete this sentence, no need to mention this twice, at the end of the paragraph (lines 205-208) is sufficient

3.0 Results 3.1 Clustering of Samples according to ANOSIM Global Test Scores p. 7 line 215: A total of 74,690 high quality bacterial sequences....

3.2 Shifts in bacterial community structure p. 7 line 223-224: Delete this sentence, it's redundant p. 7 line 230: August 2016 (dry season) samples p. 7 line 231: September 2017 (wet season) samples p. 7 line 231: There were clear overlaps between samples from... p. 7 line 232-233: We also observed a gradual shift in bacterial composition

from mineral soils and freshwater peat towards brackish and marine samples.

3.3 Bacterial Distribution according to source type and cruise p. 8 line 240: Delete "Fig 3 show that"

3.4 Alpha Diversity Indices p. 8 line 263-264: Rewrite the sentence p. 8 line 265: microbial communities varied significantly along the different source types... p. 8 line 266: ... to be higher than that of March 2017... p. 9 line 276: Authors are referring to "upstream" samples in this sentence, which samples are these? They did not clearly differentiate between upstream and downstream samples in the Methodology section which is causing confusion in subsequent text.

3.5 Functional Profile of Bacterial Communities p. 9 line 289-290: Potential KEGG pathways between (i) marine and brackish peat, and (ii) freshwater peat and mineral soil were similar. There were differences between source types and seasons p. 9 line 290-292: Delete this sentence. It's part of Discussion

3.6 Distance-based Linear Model of bacterial communities and environmental parameters p. 9 line 301: Dissolved Inorganic Phosphate (10.57%). p. 9 line 304: Delete "lastly" p. 9 line 305: Dissolved Inorganic Nitrogen (4.29%) respectively made up the... p. 10 line 308-309: Move this sentence to p. 9 line 300: "Marginal DistLM was performed in order to gauge the extent of physicochemical parameters or environmental variables accounting for a compelling proportion of variation in the bacterial communities. Significant vectors of environmental variables ($R2>0.3892$, $P <0.001$) were 308 calculated based on a linear model (DistLM) and plotted against the bacterial community composition as shown in Fig 7. Salinity was the single best predictor variable.... " p. 10 line 311-320: The distLM model clustered samples from the August 2016 cruise separately from the March 2017 samples. Brackish peat, as well as marine samples from August 2016, correlated more strongly with salinity, irrespective of land use. On the contrary, the March 2017 samples were found to cluster separately with DO. In addition, the August 2016 mineral soil samples correlated with silicate.

[Figure]

4.0 Discussion p. 10 line 332: Delete this subheading and move subheading 4.2 to 4.1 p. 10 line 335-342: in varying abundances, indicating high variation within the system. The majority of bacterial taxa were restricted to a relatively small number of assemblages. However, due to the heterogeneity of the Rajang River, substantial shifts in OTU diversity were shown, while exhibiting successional changes in community composition downstream. We observed abrupt shifts in terms of richness and diversity as well as bacterial distribution, which was structured according to macro-scale source types. Staley et al. (2015) proposed that variability in microbial communities were less due to the presence/absence but likely due to shifts in relative abundance of OTUs. p. 10 line 342: community composition, overlap between the core microbiome (i.e. free-living and particle-attached portions) of samples were not evident. p. 11 line 346: Change "further supported" with "demonstrated" p. 11 line 351: The short residence time in the Rajang River likely reflected a similar scenario to San Francisco Bay (Reference). p. 11 line 372-378: Delete these sentences. Beta-proteobacteria was already mentioned in the previous paragraph. p. 11 line 380: Were there really Cyanobacterial blooms? p. 12 line 385: Sphingomonas, a purple-sulfur bacteria, p. 12 line 391: indicating its preference for this environment. It's interesting to note that most studies on... p. 12 line 394: In most of these studies, Deinococcus-Thermus was found in low abundance (e.g. 1% in Antarctic marine environments, 1.5% in hypersaline soils; Giudice and Azzaro, 2019; Vera-Gargallo et al., 2019) when compared to the Rajang River. p. 12 line 397: Start new paragraph with: "There was a fundamental shift in bacterial community composition when taking the major taxa into consideration. There was a clear distinction between dry and wet seasons with an overall higher species richness and diversity for the dry season" For the wet season, focus on both the March and September cruises to make a conclusion

4.3 Factors determining bacterial community composition p. 13 line 421-427: Delete these sentences, was already mentioned in Methodology p. 13 line 427: There was a continual shift... p. 13 line 432: similar to findings by... p. 13 line 434: likely explaining the reduced relative abundances of some taxa. For example, Chloroflexi has a

higher relative abundance upstream while Deinococcus-Thermus shows lower relative abundance downstream. p. 13 line 438: Delete "salinity gradients" p. 13 line 451: Salinity, DIP () and dissolved oxygen are major environmental drivers of species distribution (References). In this study, marine and brackish peat samples correlated well with salinity. p. 14 line 459-469: Not sure what the authors want to say here. Do they assume there was high or low bacterial productivity? p. 14 line 478-480: While the development of unique community structures was strongly influenced by spatial factors, seasonality also played a role. Seasonal variability was also observed between the... p. 14 line 485-490: Again, can the term Cyanobacterial bloom be accurately used? p. 15 line 494: "were similar in terms of climate...."

4.4 Possible pathogenic bacteria and/or anthropogenic influence and land-use change p. 15: Start the paragraph with line 515: "The results obtained from this study suggest that the run-off from anthropogenic activities alters the microbial community composition. Anthropogenic disturbances, in particular settlements and logging (secondary forest), led to higher diversity indices (Fig .6). On the contrary, sites surrounded by oil.. ...

5.0 Conclusion p. 16 line 543: The authors refer to "pristine and less pristine environments". Which sites were classified as pristine, and which were less pristine? p. 16 line 545: The PICRUST results showed differences between source types p. 16 line 550: mixing experiments. This approach will contribute towards a better understanding of the response of microbial communities to anthropogenic perturbations, as well as their role in degrading peat-related run-off from. . .

---

## Referee Comment (RC2) · Anonymous Referee #2 · 16 Jul 2019

The manuscript of Sia et al. describes a study of bacterial communities distribution in a section of the Rajang River. Overall, the quality and content of the paper is in line with similar publications on lotic bacterial communities, where the community composition is linked to environmental parameters. The strongest point of the study is that is covers multiple time points (different seasons) and several salinity zones. The authors also made an attempt to estimate potential functions of the bacterial communities. I would like to note a detailed and comprehensive Discussion section. However, some revision is necessary. Certain results need to be verified, methods described more in details (please see specific comments). English language could be improved; the manuscript is not free of mistakes and misprints. Some specific questions and comments: P 5

[Figure]

L 146 – it is not clear for me how is classification into freshwater and brackish water described in Fig. 1(B). Possibly that is due to the poor quality of the map. P 5 L 150, 152 – Are you sure that those were polycarbonate filters? GF are usually glass fiber filters. P 5 L 156 – Incorrect reference. Caporaso et al. 2012 describe QIIME pipeline, not Illumina sequencing. P 5 L 156 – Could you please add more information on DNA extraction and library preparation procedures, for example, which primers were used for amplification? P 6 L 163 – Reference for Mothur pipeline missing. P 6 L 175 – Reference for the GreenGenes database missing. P 7 L 215 – Can you explain why the sequencing depth was so low, especially for some samples? Was it on purpose? P 7 L 215 – Were the sequences deposited to a public database? P 7 L 232 – Are you sure it is "brackish peat" and not "freshwater peat", which seems to me from Fig.2? P 8 L 247-249 – This observation is not obvious to me from Fig. 3. P 8 L 258-259 – was the difference between OTU counts statistically significant? P 10 L 324 – I didn't find any description of the results separately for free-living and particle-attached bacteria, however you discuss them a bit in chapter 4.1 in relation to Supp. Fig. 3. Were the results pooled together for free-living and particle-attached bacteria in Fig. 2-7? P 11 L 378-380 – How does the dominance of Proteobacteria indicate its role in nitrogen cycling? Please explain how it is complementary to Cyanobacteria bloom, the message is unclear. P 12 L 394- 397 – "In contrast, most extreme environments show. . ." this sentence sounds strange and needs to be rephrased.
* * *

---

## Author Comment (AC1) · 2 Sep 2019

We would like to express our gratitude to Ref #1 for the detailed comments and suggestions which helped to improve the manuscript significantly. Our point-by-point responses are posted below, with the reviewer's comments being quoted first and our response (R) below each comment.

A) General Comments This study addresses the underlying factors that may influence the spatial and seasonal distribution of the prokaryotic communities and nutrient dynamics along the Rajang River, South China. Although the results of this study are valid and interesting, there are several points that need to be addressed.

[Figure]

1) Dividing sampling cruises into "wet season" and "dry season" may be more beneficial than referring to them individually. Authors mentioned both wet and dry seasons in the Methodology section (section 2.1), however, sampling cruises associated with each are lacking.

R: We agree that it would be beneficial to classify the sampling cruises into "dry season" and "wet season", however as the two "wet" seasons also differ in terms of its microbial community composition, we kept the individual cruises in order to prevent confusion between the two wet seasons. We have clarified the 'classification' of the three cruises as wet or dry season in the method section (it now reads: The August 2016 cruise (colored red) is classified as the dry season based on the lower mean rainfall value as compared to the other two (March 2017 and September 2017), in which the both are classified as the wet season (refer to Sup. Fig. 1).

2) The site map (Figure 1A) currently shows sampling points throughout the river with source types (Figure 1B), but illustration of anthropogenic activities along the river is missing. It will be helpful to add these as it's not clear which sections of the river are impacted by which activities.

R: Thank you for this. For the anthropogenic activities, the data was extracted from a report done by Wetlands International (2015) and is more a qualitative description. This description was then used for the classification of land use. The Fig 1(B) was intended only to show the zones of peatlands and not for the anthropogenic activities.

3) Additional statistical analysis, such as PERMANOVA, may be used to infer the impact of anthropogenic activities (e.g. human settlements,effluents, transportation and sand dredging) and source types on beta diversity. Much of the Results and Discussion sections revolve around alpha diversity indices but very little is mentioned about beta diversity.

R: Thank you for pointing this out. Beta diversity was in fact calculated and used for the discussion, however, obviously, not clearly pointed out. For example, the plotting

of nMDS via PRIMER includes Kruskal-Wallis calculations (Kruskal stress formula: 1; Minimum stress: 0.01; 2-d: Minimum stress 0.18 occurred 21 times). Furthermore, the resemblance matrix was calculated using the Bray-Curtis dissimilarity measure. We have included this information in the methods section to reflect its inclusion in our analyses.

4) Potential functionality inferred from PICRUST showed clear distinction between samples when comparing source type. It would be interesting to see if potential functionality differed too such an extent when samples are compared by anthropogenic perturbations.

R: Thank you for the suggestion. While comparing the potential functionality of the anthropogenic perturbations, there was not much variation across the different anthropogenic activities, hence this was not included in the results and discussion.

5) The general flow of the Discussion section needs improvement. The significance and contribution of the study will have a bigger impact when the Discussion is presented clear and logically. Also, the authors should double check the tense (present/past/passive) for each section.

R: Thank you for pointing this out. We have rearranged the discussion section whereby the bulk of the discussion was the drivers of microbial community composition and was separated into 3 sections, i.e. spatial and environmental drivers, seasonal drivers and anthropogenic drivers.

6) Recheck format of in-text references. Not all citations are written in the same format.

R: Thank you. We have checked through the in-text reference and changed those that have errors.

B) Specific Comments

1.0 Introduction p. 4 paragraph 6: The authors aimed to study microbial diversity and potential function in the Rajang River. Although this study is the first to investigate

microbial diversity along a freshwater-marine gradient, with a tropical peatland component, the importance of the river in Malaysia and clear objectives need to stated.

R: Thank you for highlighting this. The last paragraph of the introduction was combined with the last few sentences from the previous paragraph. This paragraph now reads: "This study focuses on the Rajang River, which is the longest river in Malaysia and one of the most socio-economically important peat-draining rivers in South East Asia. It transports large amounts of terrestrial material (Müller-Dum et al., 2019) experiences two monsoonal seasons (Sa'adi et al., 2017) and is subjected to anthropogenic disturbances (Gaveau et al., 2016; Miettinen et al., 2016). Thus, it is fundamental to take into consideration both seasonal and anthropogenic influences on the microbial communities of the Rajang River. Given the rapid development in Sarawak and the importance of microbes in several biogeochemical processes in the Rajang river (Jiang et al., 2019; Martin et al., 2018; Müller-Dum et al., 2019; Zhu et al. 2019), it is imperative to study the microbial communities to enable future predictions and management responses. The Rajang river offers the opportunity to study the microbial diversity along a river to sea continuum and at the same time assess influence of natural conditions such as seasons (dry vs. wet), different soil types (peat vs. mineral soil), as well as anthropogenic disturbances (e.g human settlements and plantations) on microbial succession. This study aims to investigate (1) the microbial community structure, diversity and probable function across wet and dry seasons in order to (2) understand the underlying factors that may influence the spatial and seasonal distribution of the prokaryotic communities and the nutrient dynamics involved in the Rajang River."

2.0 Methodology 2.1 Study area and sampling strategy p. 5 line 136-139: "According to Wetlands International (2015), the land surrounding the study sites is characterized by a range of anthropogenic activities, ranging from oil palm and sago plantations to human settlements as well as transportation and sand dredging activities (Fig. 1(B))." This is not clear from Fig 1B. Colours are associated with forested or non-forested lands, however, the map does not depict the different anthropogenic activities along

the river.

R: Thank you for this. For the anthropogenic activities, the data was extracted from a report done by Wetlands International (2015) and is more a qualitative description. This description was then used for the classification of land use. The Fig 1(B) was intended only to show the zones of peatlands and not for the anthropogenic activities. We will create a new supplementary figure highlighting the anthropogenic activities although it will have to be based on mostly informal data due to the lack of official statistics or other related publications.

2.1 Pyrosequencing and Bioinformatics Analyses. Change "Pyrosequencing" to "Illumina sequencing" in the subheading. The authors did not perform 454-pyrosequencing but Illumina sequencing. Did ACE also extract DNA from samples? If that's the case, the authors should mention this in the beginning of this section.

R: Agreed. "Pyrosequencing" was changed to "Illumina sequencing". Yes, ACE also extracted the DNA samples. This information was placed at the sentence before section 2.2.

3.1 2.4 Statistical Analyses and distLM model. The authors used db-RDA to determine the impact of environmental variables on microbial diversity. The same method can be used to determine which parameters have an influence on specific bacterial taxa. Likewise, Spearman/Pearson correlations can be drawn between environmental parameters and taxa. The information inferred from these additional analyses can help the authors to link certain taxa to specific source types or pollution sources.

R: We agree that this would be a good addition to the existing discussion and will carry out the suggested analyses. Spearman's ranking on the major taxa does support the key role of salinity shaping the microbial diversity.

I also suggest that anthropogenic inputs should be divided into the following categories: human settlements, effluents (from both palm oil and sago plantations), transportation

and sand dredging. Variation partitioning, if possible, may then be used to determine which anthropogenic input, or source type, had the biggest impact on bacterial diversity along the river.

R: Thank you for this suggestion. We did indeed use variation partitioning for the distLM models. Unfortunately we do not think that we have sufficient data points from areas affected by sand dredging to be included in the model. Transportation of logs and sand by boats can be observed throughout the whole river, making it difficult to distinguish its impact between different sites.

3.0 Results

3.3 Bacterial Distribution according to source type and cruise This section may be improved by organizing it into the following paragraphs: Mention the dominant taxa and their relative abundances. The author mentioned this in the Discussion section (section 4.1 line 333-335) but not in the Results section.

R: Agreed, this was moved to the results section 3.3.

Which taxa (dominant or specialized) were more predominant at specific source types and/or seasons? In this section, the authors acknowledge a higher Cyanobacterial abundance for the September 2017 marine and brackish peat samples. In the Discussion section they refer to the higher Cyanobacterial counts as "blooms" without prior testing (e.g. chlorophyll-a) as a proxy measure to confirm algal blooms. Since Cyanobacteria are more pronounced during warmer months, and anthropogenic activities close to the sampling areas can cause nutrient input and thus proliferation of Cyanobacteria, how valid is the assumption/statement made in the manuscript without additional measurements?

R: Thank you for pointing this out. There were two measurements for validating cyanobacterial abundance. One was the respiration experiment, which showed that there was greater respiration as compared to oxygen production. The other was phytoplankton identification via pigments using a software (CHEMTAX). Only two sampling cruises were available for the phytoplankton identification, which unfortunately for September 2017 is unavailable. However, between the dry (August 2016) and wet (March 2017) seasons, the wet season did indeed show greater counts of Cyanobacteria.

Alpha Diversity Indices: How did the authors calculate the effect of land use and source type on alpha diversity indices? This was not mentioned in the Methodology section.

R: The information "The alpha diversity was calculated using the estimate_richness function embedded within the plot_richness function found within the phyloseq package utilizing R (v.3.5.3)." is now included in Section 2.2

Instead of comparing indices between cruises, the authors can make comparisons between seasons (e.g. compare the entire wet season with the dry season).

R: Referring to the earlier explanation, we kept the individual cruises as the two wet seasons exhibited different microbial community composition, which warrants two separate cruises instead of the entire wet season.

Discussion: I recommend that the authors re-write certain paragraphs of the Discussion section so that it may have a bigger impact on readers. Instead of naming all the different types of taxa in the river, focus on the important ones and what their roles are. How does the environment and different inputs (source types and anthropogenic activities) impact these taxa?

R: Thank you for highlighting this. The sections were further subdivided whereby the main discussion was focused on the factors determining bacterial community composition and was subdivided into three components which are (4.2.1 Spatial and environmental drivers, 4.2.2 Seasonality as a driver of microbial community composition and 4.2.3 Land-use change and anthropogenic drivers.

Possible pathogenic bacteria and/or anthropogenic influence and land-use change.

Was Flavobacterium the only potential pathogen identified? I would suggest to start the paragraph with anthropogenic influence and land-use change. A second, shorter paragraph can discuss the potential pathogens

R: The possible pathogenic bacteria identified were part of the CFB group. We grouped together the other information regarding Proteobacteria and Bacteroidetes as part of the paragraph relating to land use change and anthropogenic drivers.

C) Technical comments

1.0 Introduction p. 3: Combine paragraphs 2 and 5. Both are discussing lotic environments and nutrient cycling; it will thus make more sense to combine these two.

Response: Agreed, the two paragraphs were combined. This paragraph now reads: "Lotic environments are the interface between soil and aquatic environments and aquatic environments as terrestrial environments seed microbes into the adjacent water column due to flowing waters (Crump et al., 2012). Until not long ago, rivers were thought to be passive channels in the global and regional determination of carbon (C) and weathering products until it became clear that rivers regulate for example the transfer of nutrients from land to coastal areas (Smith and Hollibaugh, 1993). Several studies have shown that bacteria are key players in nutrient processing in freshwater systems (Cotner and Biddanda, 2002; Findlay, 2010; Madsen, 2011). Zhang et al. (2018a) stated that the organic matter composition is strongly modified by bacteria as well as its resistance to degradation. Bacteria strongly influence the fluvial organic matter, hence playing a role in carbon cycle (Dittmar et al., 2001) and recent studies in the Rajang river have demonstrated that as indicated by high concentrations of D-form amino acids (Zhu et al., 2019). Moreover, it was demonstrated by Jiang et al. (2019) that Dissolved Organic Nitrogen was reduced to $NH_4^+$ via mineralization and ammonification, again highlighting the biogeochemical activity and the importance of microbes in the Rajang River. Until now, there has, however, been no study on their diversity yet; a gap that this study aims to fill. Thus, it is essential to understand the dynamics and

structure of microbial communities in them to assess their contribution towards bio-geochemical fluxes such as carbon and nitrogen (Battin et al., 2008; Raymond et al., 2013), as well as phosphate cycling (Hall et al., 2013). In addition, the fluxes as well as transformations of organic matter as well as nutrients in aquatic systems are environmentally driven by parameters such as temperature or the availability of nutrients in these ecosystems (Welti et al., 2017). In turn, various gradients (i.e physical, chemical, hydrological or even biological) contribute to the changes in the microbial diversity and distribution living within the lotic environments (Zeglin, 2015)."

p. 4 line 93-93: Due to their high diversity and fast generation time, microbial communities are the first responders to environmental changes

R: Agreed and changed to recommended sentence.

p. 4 line 96: Liao et al. (2019) showed that p. 4 line 97: delete "further" p. 4 line 97: Bruland et al. (2008) demonstrated that the

R: Agreed, removed "further"

p. 4 line 99-102: "Thus, as the Rajang River experiences two monsoonal seasons (Sa'adi et al., 2017) and is subjected to anthropogenic disturbances (Gaveau et al., 2016; Miettinen et al., 2016), it is thus fundamental to take into consideration both seasonal and anthropogenic influences on the microbial communities of the Rajang River." This forms part of the aim and objective and should rather move to last paragraph

R: Agreed, this was changed to better reflect the aim and importance of the Rajang River.

p. 4 line 115: delete "hypothesized"

R: Agreed, removed "hypothesized"

p. 4 line 120-121: as well as anthropogenic disturbances (e.g. human settlements and plantations) on microbial succession.

R: Agreed, changed to recommended sentence.

p. 4 line 121-122: Delete "Linear models are used to examine the relationship between the microbial community structure and their environment."

R: Agreed to remove sentence as it was already explained in methodology.

2.0 Methodology 2.1 Study area and sampling strategy p. 5 line 130: The region:

R: Agreed. Changed to "The"

p.5 line 134: small tributaries

R: Agreed, changed from "distributaries" to "tributaries"

p. 5 line 142: Which months were associated with the wet and dry season, respectively?

R: The following sentence was extracted from the caption of Sup. Fig. 1. " The August 2016 cruise (colored red) is classified as the dry season based on the lower mean rainfall value as compared to the other two (March 2017 and September 2017), in which the both are classified as the wet season." to be placed in-text in the methodology for ease of reference

p. 5 line 149: Approximately 250 – 500 mL of water: : :.

R: Agreed, changed to "approximately"

p. 5 line 153-156: A total of 117 filters were recovered (1 x 3.0 _m filter was discarded due to contamination) and immediately stored at - 20_C.

R: Agreed. The sentence was changed as recommended.

2.2 Pyrosequencing and Bioinformatics Analyses p. 5 line 160: Briefly, fastq files generated: : :

R: Agreed. "In short" changed to "Briefly"

p. 6 line 161: quality trimmed with fastqc, primer sequeces: : :.

R: Agreed, changed from "processed" to "quality trimmed"

p. 6 line 162-163: High quality sequences were subsequently processed using the Mothur pipeline.

R: Agreed. Changed to the recommended sentence.

p. 6 line 164: SILVA database

R: Agreed, "alignment" changed to "database"

p. 6 line 171: potential functional genes

R: Agreed, added "potential"

2.3 Physico-chemical Data and Geochemical Analyses p. 6 line 179: in-between the Cruises

R: Agreed, added "-"

p. 6 line 189-191: Belawai samples (2_13'47.16"N, 111_12'19.04"E) were used in an incubation experiment to study the net primary productivity and respiration rate of the Rajang River. Technical triplicates were incubated in both light and dark set-ups (Refer to Supp. Table 1 for details).

R: Agreed, the sentence was modified to the recommended sentence.

2.4 Statistical Analyses and distLM model p. 6 line 195-197: to determine if the various terrestrial source types or different land use impacted bacterial community composition.

R: Agreed, sentence structure was changed to the recommended.

p. 7 line 199: what type of normalization method was used?

R: The following sentence was added: "using the "Normalise Variables" function in the

PRIMER 7 software".

p. 7 line 202-204: "The authors would like to note that the distLM models are based on only the August 2016 and March 2017 cruise as there was a lack of physico-chemical data from the September 2017 cruise due to malfunctioning equipment." Delete this sentence, no need to mention this twice, at the end of the paragraph (lines 205-208) is sufficient

R: Agreed, the sentence was removed.

p. 7 line 215: A total of 74,690 high quality bacterial sequences: : :.

R: Agreed and changed as recommended.

3.2 Shifts in bacterial community structure p. 7 line 223-224: Delete this sentence, it's Redundant

R: Agreed, the sentence was removed.

p. 7 line 230: August 2016 (dry season) samples

R: Agreed, added "(dry season)" to the sentence.

p. 7 line 231: September 2017 (wet season) samples

R: Agreed, added "(wet season)" to the sentence.

p. 7 line 231: There were clear overlaps between samples from:

R: Agreed, changed from "there are apparent" to "there were clear"

p. 7 line 232-233: We also observed a gradual shift in bacterial composition from mineral soils and freshwater peat towards brackish and marine samples.

R: Agreed, sentence was changed accordingly as recommended.

3.3 Bacterial Distribution according to source type and cruise p. 8 line 240: Delete "Fig 3 show that"

R: Removed as recommended but added (Fig. 3) to the end of the sentence.

3.4 Alpha Diversity Indices p. 8 line 263-264: Rewrite the sentence

R: Sentence was rewritten as "For the September 17 cruise, we observed increased values of Chao1 across the brackish peat, freshwater peat as well as mineral soils."

p. 8 line 265: microbial communities varied significantly along the different source types

R: Agreed, changed sentence as recommended.

p. 8 line 266: to be higher than that of March 2017:

R: Agreed, changed from "found in" to "of"

p. 9 line 276: Authors are referring to "upstream" samples in this sentence, which samples are these? They did not clearly differentiate between upstream and downstream samples in the Methodology section which is causing confusion in subsequent text.

R: Agreed. Added explanation at the end of the text: (i.e. Human Settlement, Oil Palm and Sago Plantation, Oil Palm Plantation and Secondary Forest).

p. 9 line 289-290: Potential KEGG pathways between (i) marine and brackish peat, and (ii) freshwater peat and mineral soil were similar. There were differences between source types and seasons

R: Agreed, the recommended sentence provided more clarity.

p. 9 line 290-292: Delete this sentence. It's part of Discussion

R: Agreed and removed.

p. 9 line 301: Dissolved Inorganic Phosphate (10.57%)

R: removed "at" and added parenthesis to "10.57%".

p. 9 line 304: Delete "lastly"

R: Agreed.

p. 9 line 305: Dissolved Inorganic Nitrogen (4.29%) respectively made up the

R: Agreed, changed from "(4.29%, respectively)" to "(4.29%) respectively

p. 10 line 308-309: Move this sentence to p. 9 line 300: "Marginal DistLM was performed in order to gauge the extent of physicochemical parameters or environmental variables accounting for a compelling proportion of variation in the bacterial communities. Significant vectors of environmental variables (R2>0.3892, P <0.001) were calculated based on a linear model (DistLM) and plotted against the bacterial community composition as shown in Fig 7. Salinity was the single best predictor variable

R: Agreed and changed.

p. 10 line 311-320: The distLM model clustered samples from the August 2016 cruise separately from the March 2017 samples. Brackish peat, as well as marine samples from August 2016, correlated more strongly with salinity, irrespective of land use. On the contrary, the March 2017 samples were found to cluster separately with DO. In addition, the August 2016 mineral soil samples correlated with silicate.

R: Agreed and changed to suggested sentences.

p. 10 line 332: Delete this subheading and move subheading 4.2 to 4.1

R: Agreed, the remaining labels were corrected accordingly.

p. 10 line 335-342: in varying abundances, indicating high variation within the system. The majority of bacterial taxa were restricted to a relatively small number of assemblages. However, due to the heterogeneity of the Rajang River, substantial shifts in OTU diversity were shown, while exhibiting successional changes in community composition downstream. We observed abrupt shifts in terms of richness and diversity as well as bacterial distribution, which was structured according to macro-scale source types. Staley et al. (2015) proposed that variability in microbial communities were less

due to the presence/absence but likely due to shifts in relative abundance of OTUs.

R: Agreed and changed.

p. 10 line 342: community composition, overlap between the core microbiome (i.e. free-living and particle-attached portions) of samples were not evident.

R: Agreed and changed.

p. 11 line 346: Change "further supported" with "demonstrated"

R: Agreed and changed.

p. 11 line 351: The short residence time in the Rajang River likely reflected a similar scenario to San Francisco Bay (Reference).

R: Agreed and changed.

p. 11 line 372-378: Delete these sentences. Beta-proteobacteria was already mentioned in the previous paragraph.

R: Agreed and removed.

p. 11 line 380: Were there really Cyanobacterial blooms?

R: Thank you for pointing this out. Cyanobacterial bloom was changed to "the higher abundance of Cyanobacteria", which more accurately describes the composition as shown by the abundance in taxa.

p. 12 line 385: Sphingomonas, a purple-sulfur bacteria,

R: Agreed and changed.

p. 12 line 391: indicating its preference for this environment. It's interesting to note that most studies on

R: Agreed and changed to recommended sentence.

p. 12 line 394: In most of these studies, Deinococcus-Thermus was found in low abundance (e.g. 1% in Antarctic marine environments, 1.5% in hypersaline soils; Giudice and Azzaro, 2019; Vera-Gargallo et al., 2019) when compared to the Rajang River.

R: Agreed and changed.

p. 12 line 397: Start new paragraph with: "There was a fundamental shift in bacterial community composition when taking the major taxa into consideration. There was a clear distinction between dry and wet seasons with an overall higher species richness and diversity for the dry season" For the wet season, focus on both the March and September cruises to make a conclusion

R: Agreed and changed.

p. 13 line 421-427: Delete these sentences, was already mentioned in Methodology

R: Agreed, the sentences were removed.

p. 13 line 427: There was a continual shift

R: Agreed, changed "is" with "was".

p. 13 line 432: similar to findings by

R: Agreed, changed "akin" to "similar"

p. 13 line 434: likely explaining the reduced relative abundances of some taxa. For example, Chloroflexi has a higher relative abundance upstream while Deinococcus-Thermus shows lower relative abundance downstream.

R: Agreed and changed to recommended sentence.

p. 13 line 438: Delete "salinity gradients"

R: Agreed and removed.

p. 13 line 451: Salinity, DIP () and dissolved oxygen are major environmental drivers

of species distribution (References). In this study, marine and brackish peat samples correlated well with salinity.

R: Agreed and changed.

p. 14 line 459-469: Not sure what the authors want to say here. Do they assume there was high or low bacterial productivity?

R: We deduced that even though there was high abundance of associated phyla that may contribute to the production of O2 (via primary production), the high CO2 emissions and higher respiratory rate show that there was higher bacterial productivity versus primary production.

p. 14 line 478-480: While the development of unique community structures was strongly influenced by spatial factors, seasonality also played a role. Seasonal variability was also observed between the

R: Agreed and changed to recommended sentence.

p. 14 line 485-490: Again, can the term Cyanobacterial bloom be accurately used?

R: Thank you once again for pointing this out. The sentence was changed to "The greater abundance of Bacteroidetes in March 2017 may be indicative of the community composition adjusting due to the processing of organic material caused by the higher cyanobacterial abundance in the September 2017 cruise. This was similar to a study by Pinhassi et al., (2004), in which the higher abundance of Bacteroidetes follows after an algal bloom.". This would reduce the assumption of a cyanobacterial bloom. The study quoted (Pinhassi et al. (2004)) was used as an example for probable inference of cyanobacterial bloom but cannot yet be confirmed.

p.15 line 494: "were similar in terms of climate"

R: Agreed, and changed.

p. 15: Start the paragraph with line 515: "The results obtained from this study suggest

**BGD**

that the run-off from anthropogenic activities alters the microbial community composition. Anthropogenic disturbances, in particular settlements and logging (secondary forest), led to higher diversity indices (Fig .6). On the contrary, sites surrounded by oil

R: Agreed and changed. The breaking of paragraphs provide better clarity to the overall flow.

p. 16 line 543: The authors refer to "pristine and less pristine environments". Which sites were classified as pristine, and which were less pristine?

R: Thanks for pointing this out. We have changed this to "anthropogenic perturbations (regions with oil palm plantations and human settlements) led to increased richness but less diversity compared to those that were less affected by anthropogenic perturbations (coastal zone and secondary forest)."

p. 16 line 545: The PICRUST results showed differences between source types

R: Agreed, changed "difference" to "differences"

p. 16 line 550: mixing experiments. This approach will contribute towards a better understanding of the response of microbial communities to anthropogenic perturbations, as well as their role in degrading peat-related run-off from

R: Agreed, and changed to suggested sentence.

––––––––––––––––––––––––

---

## Author Comment (AC2) · 2 Sep 2019

We would like to thank Ref #2 for the comments and suggestions which helped to improve the manuscript significantly. Our point-by-point responses are posted below, with the reviewer's comments being quoted first and our response (R) below each comment.

The manuscript of Sia et al. describes a study of bacterial communities' distribution in a section of the Rajang River. Overall, the quality and content of the paper is in line with similar publications on lotic bacterial communities, where the community composition is linked to environmental parameters. The strongest point of the study is that is covers

multiple time points (different seasons) and several salinity zones. The authors also made an attempt to estimate potential functions of the bacterial communities. I would like to note a detailed and comprehensive Discussion section. However, some revision is necessary. Certain results need to be verified, methods described more in details (please see specific comments). English language could be improved; the manuscript is not free of mistakes and misprints.

Some specific questions and comments:

P 5 L 146 – it is not clear for me how is classification into freshwater and brackish water described in Fig. 1(B). Possibly that is due to the poor quality of the map.

R: Thank you for pointing this out. We removed this sentence "as described in Fig 1. (B)" as Fig. 1(B) is to show the areas with peat only.

P 5 L 150, 152 – Are you sure that those were polycarbonate filters? GF are usually glass fiber filters.

R: Thank you for pointing this out. The correct filter used was Nuclepore™ Track-Etched Polycarbonate Membrane Filter. We have removed the (GF/C) description.

P 5 L 156 – Incorrect reference. Caporaso et al. 2012 describe QIIME pipeline, not Illumina sequencing.

R: Agreed. Changed to Bentley et al. (2008) which describes the first paper that Illumina was based upon.

P 5 L 156 – Could you please add more information on DNA extraction and library preparation procedures, for example, which primers were used for amplification?

R: Thank you for pointing this out. We have included the relevant information in the methods section. It now reads: ....A total of 117 filters were recovered (1 x 3.0 $\mu$m was discarded due to contamination) and immediately stored at -20 °C and sent to the Australian Centre for Ecogenomics (ACE), Brisbane for DNA extraction, library preparation and processing utilizing the Illumina (Bentley et al., 2008) platform.

2.2 Illumina Sequencing and Bioinformatics Analyses Initial upstream processes were carried out by the Australian Centre for Ecogenomics utilizing the ACE mitag pipeline (ACE, 2016). The primers utilized were based on the V3 – V4 hypervariable regions of the 16S rRNA gene.

P 6 L 163 – Reference for Mothur pipeline missing.

R: Thank you for pointing this out. The relevant citation was added (Schloss et al., 2009)

P 6 L 175 –Reference for the GreenGenes database missing.

R: Thank you for pointing this out. The relevant citation was added (DeSantis et al., 2006)

P 7 L 215 – Can you explain why the sequencing depth was so low, especially for some samples? Was it on purpose?

R: Thank you for this question. The minimum sequencing depth was 10,000 reads per sample. After QC and removal of unknown sequences, some samples were left with a very low read count. Given the general lack of data from these systems and to 'lose' as little information /samples as possible, we chose a low read number for the subsampling.

P 7 L 215 – Were the sequences deposited to a public database?

R: As of now, the sequences have yet to be deposited in a public database. They will be submitted in the coming days.

P 7 L 232 – Are you sure it is "brackish peat" and not "freshwater peat", which seems to me from Fig.2?

R: Yes, thank you for pointing this out. "Brackish peat" was changed to "freshwater

peat"

P 8 L 247-249 – This observation is not obvious to me from Fig. 3.

R: Agreed, this portion was removed.

P 8 L 258-259 – was the difference between OTU counts statistically significant?

R: The results shown were plotted based on the calculations from the estimate_richness function in the phyloseq package, and hence the observation was more a qualitative observation.

P 10 L 324 – I didn't find any description of the results separately for free-living and particle-attached bacteria, however you discuss them a bit in chapter 4.1 in relation to Supp. Fig. 3. Were the results pooled together for free-living and particle-attached bacteria in Fig. 2-7?

R: Thank you for pointing this out. Yes, for Figures 2 – 7 the results were pooled together for discussion as the difference between free-living and particle-attached bacteria did not exhibit clear distinction and hence was not further elaborated. The following sentence was added in Section 2.2: "Apart from the results and discussion shown for free-living and particle-attached bacteria, the remaining discussion is based on the pooled results of both components"

P 11 L 378-380 – How does the dominance of Proteobacteria indicate its role in nitrogen cycling? Please explain how it is complementary to Cyanobacteria bloom, the message is unclear.

R: The sentence was rephrased as "In a study by Yang et al. (2013), the dominance of Protebacteria influenced the nitrogen cycle via the processes of nitrification and denitrification, in which aeration would increase its abundance and result in higher mortality of cyanobacteria.

P 12 L 394- 397 – "In contrast, most extreme environments show" this sentence sounds

strange and needs to be rephrased.

R: Agreed, this sentence was changed to "In most of these studies, Deinococcus-Thermus was found in low abundance (e.g. 1% in Antarctic marine environments, 1.5% in hypersaline soils; Giudice and Azzaro, 2019; Vera-Gargallo et al., 2019) when compared to the Rajang River."